# Griffith theory of physical fractures, statistical procedures and entropy production: Rosetta stone's legacy

**Marcelo Calcina-Nogales**[1], **Boris Atenas**[2]*, **Juan Cesar Flores**[2]

**1** Carrera de Física, Facultad de Ciencias Puras y Naturales, Universidad Mayor de San Andrés, La Paz, Bolivia, **2** Departamento de Física, Facultad de Ciencias, Universidad de Tarapacá, Arica, Chile

\* batenas@academicos.uta.cl

**Data Availability Statement:** All relevant data are within the paper and its Supporting information files.

## Abstract

A physical model, based on energy balances, is proposed to describe the fractures in solid structures such as stelae, tiles, glass, and others. We applied the model to investigate the transition of the Rosetta Stone from the original state to the final state with three major fractures. We consider a statistical corner-breaking model with cutting rules. We obtain a probability distribution as a function of the area and the number of vertices. Our generic results are consistent with the current state of the Rosetta Stone and, additionally, predictions related to a fourth fracture are declared. The loss of information on such heritage pieces is considered through entropy production. The explicit quantification of this concept in information theory stays examined.

## 1 Introduction

Griffith criterion is a concept that explains the conditions under which a solid material will fracture or break. It is based on energetic considerations, specifically the balance between the energy required to create new areas (surface energy) and the bulk energy (strain energy). This criterion is directly related to this work. Explicitly, the Rosetta Stone [1–4] (Fig 1) is considered a relevant discovery in human history and, was the key to establishing a translation for Egyptian hieroglyphs. This stele was discovered inside the walls of Rosetta (Rashid) and, finally, carried up to a London museum.

In this work, we propose a model based on the use of random numbers (such as Monte Carlo) [5] to describe the fracture process of a solid material in terms of its number of vertices. Roughly speaking, we model the solid as a square polygon and propose that there are breakable points, called "breaking points", on each side of the square. A fracturing event or vertex loss starts at a random breaking point on one side of the square and propagates to another random breaking point on another adjacent side. This fracturing occurs at each vertex, reaching a final state with a few vertices ranging between 4 and 8. Next, M copies of this system (statistical ensemble) are created, which evolve individually in the same way as aforementioned. By counting the area and the number of vertices of each copy at its final state, we derive a probability distribution and use it to find the most likely polygon. As an application of the model, we use this distribution and the Griffith criterion to study the fracture process of the Rosetta stone

**Funding:** The authors received no specific funding for this work.

**Competing interests:** The authors have declared that no competing interests exist.

and conjecture about its eventual future. Finally, we identify a quantitative measurement for the missing information in the lost parts of the aforementioned stele. This task is carried out with the generic concept of entropy, and its relationship with information theory.

We believe that our model could be useful in studying tiles, glass, solar panel breakages, stelae, among others. This kind of methodology has been widely used to describe different kinds of systems; particle transport problems, chemical reactions, financial markets, social protests, higher education and theoretical models [6–20]. In addition, similar studies have been carried out in previous works [21–26] to understand the arising of cracks in paintings.

The model proposed here could be useful also to explain:

**Rolling stones in rivers**: The rounding of stones is produced by the abrasion [27] induced during the transport of these in the rivers. An analytical theory of abrasion can be found in the work of Krumbein. [28] Since then, experimentally a tumbling barrel has been used to simulate the abrasion process [29–31].

**Crack networks**: Developed in mud [32], clay [33], dry agricultural fields, master paintings [34, 35], and geological rocks [36–38]. A theory of crack formation can be found in Landau-Lifshitz [21], essentially the condition to appear crack in solid materials depends on the rigidity, stress tensor, surface area, and volume. When cracks appear they spread with vibrations [39] phenomenon that could also eventually occur in quantum circuits [40–42].

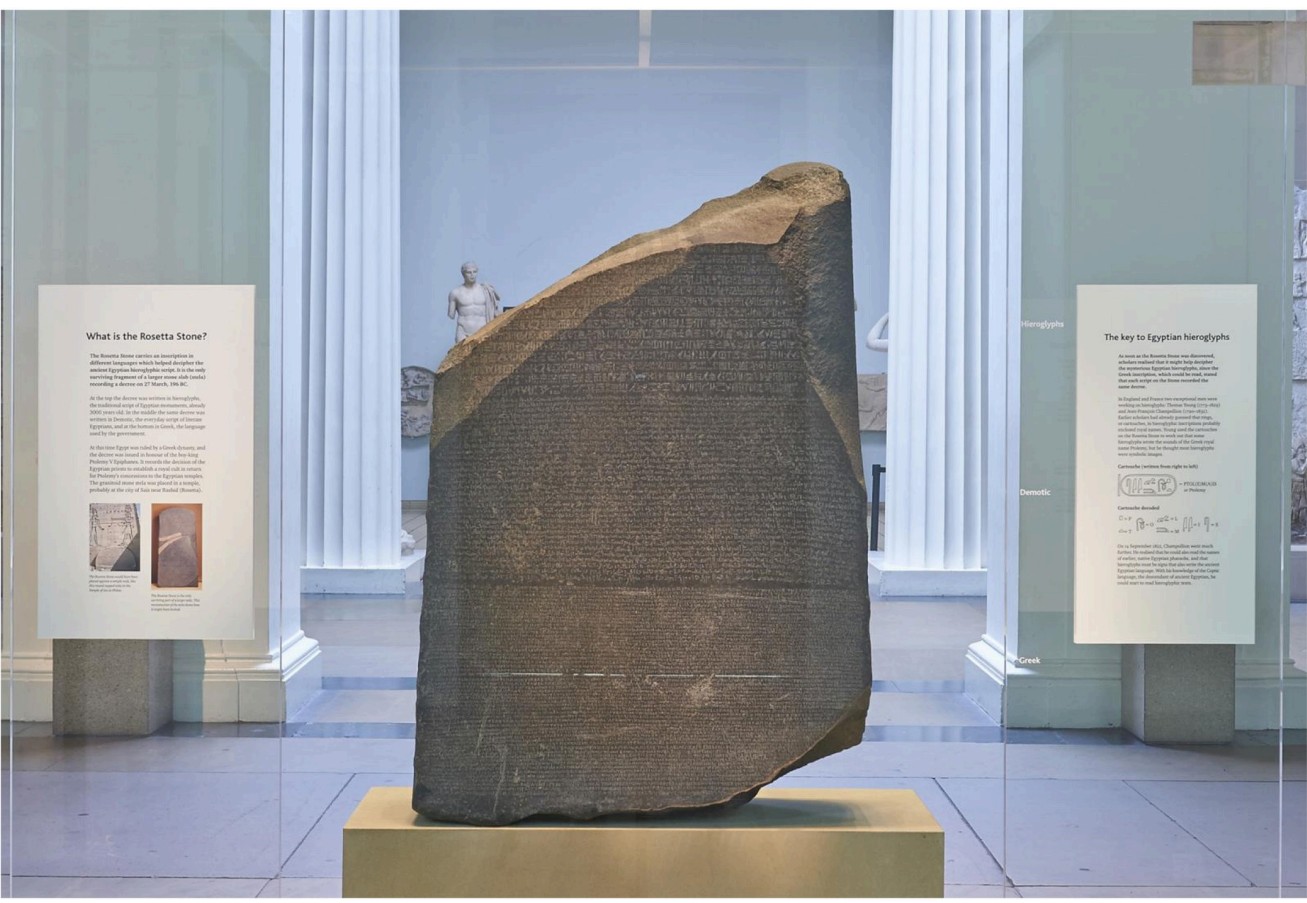

**Fig 1. The Rosetta Stone, courtesy of British Museum [1].**

**Breakage of solid structures**: At the industry, the analysis of the breaking of ceramic tiles, glass, solar panels, and others [43–57], constitutes an important issue, due to the special interest in the early detection of cracks, regarded as a serious safety concern.

This paper is organized as follows, in section 1 we present the statistical model for the process of missing corners in a plaque. In section 2, based on section II, we explicitly calculate the probabilities of vertex breaks applied to the Rosette Stone. A prediction is made regarding a possible new fracture. Section 3 considers, based on the theory of elasticity, the possible sequential history of ruptures. Furthermore, it evaluates the lost area (size) of an eventual new break. In Section 4, assuming that entropy is a measure of misinformation, a practical formula is presented for its evaluation. The production of entropy is related to the systematic loss (dispersion) of fragments of this ancient stele. Finally, in section 5 we summarize the relevant results and discuss some model limitations.

## 2 Fractures model

Let us suppose we have a solid polygon that is susceptible to cracking due to external factors like temperature changes, unexpected falls, gravity, or others. It is natural to think that the material could lose all or some of its vertices. In that case, what will the final shape of the polygon be after it has cracked? This section aims to provide an answer through numerical simulations.

### 2.1 Basic idea and numerical implementation

For the sake of simplicity, we are going to set up a model of a 2D square solid that loses its vertices. To do so, consider the following constraints. The initial solid will be modeled as a 4-side polygon that will lose its vertices randomly until it becomes another polygon with *n* vertices, which should be inscribed on the initial polygon.

As an illustrative example of the previous idea, Fig 2a shows a polygon where each side is divided into equal parts by colored points. Hereinafter, these points will be referred to as

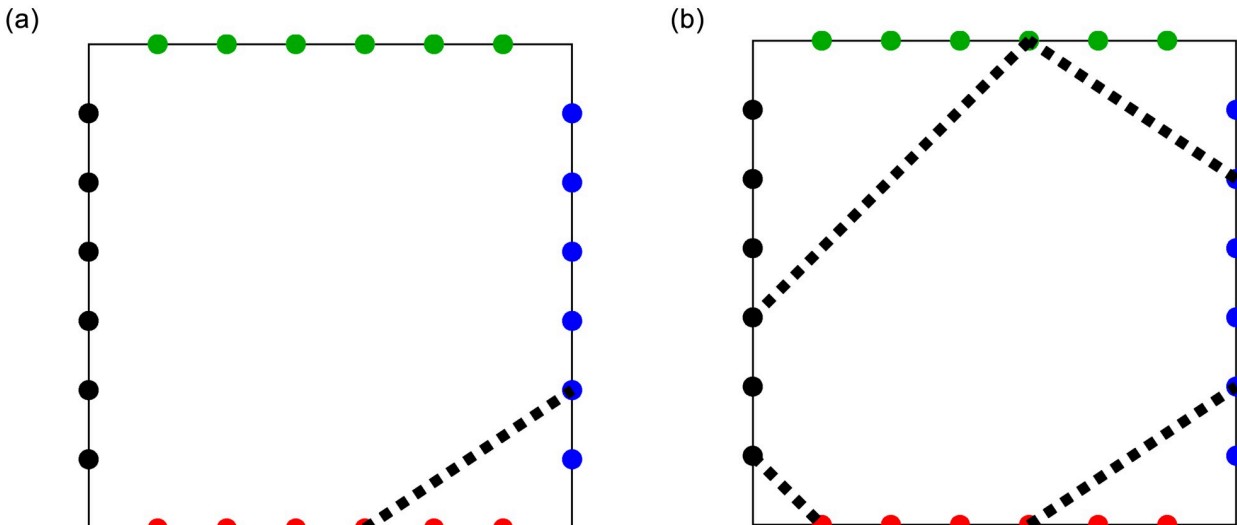

**Fig 2. Square polygon with six breaking points per side.** The colored dots divide each of the sides into seven equal parts. (a) The dotted line illustrates a fracture event for a vertex that starts at a random blue point and ends at another random red point. (b) Resulting final polygon after undergoing four fracture events.

*breaking points*, which are points with a high probability of fracturing. On the other hand, we define a fracture event as the loss of any of the vertices of the polygon. For example, the fracture event at the bottom right vertex of Fig 2a, illustrated by the dotted line, starts at a random point on the bottom side of the square and arrives at another random point on the other adjacent side. This fracturing event is repeated randomly in each of the remaining vertices of the square and that will eventually end up in a fractured square, thus obtaining a different polygon from the initial one. In the particular case of the square in Fig 2, after losing all of its vertices, the final polygon has seven vertices, as shown in Fig 2b.

Notice, in Fig 2a, the absence of breaking points at the vertices of the square. This constraint is crucial to prevent a potential situation where no actual breakage occurs due to the random selection process in our algorithm. By excluding corner breakpoints, we ensure that there will always be an adjacent side available for the algorithm to generate meaningful breaks.

## 2.2 Algorithm

To capture the basic idea described in the previous section, there will be two elements of randomness that will be considered. The first one is the random selection of a vertex in the polygon, and the second one is the random selection of breaking points on the adjacent sides of the vertex. The probability distribution of these selection events is assumed to be uniform. The algorithm that gets the final polygon can be summarized by the following sequence of steps:

- A vertex is randomly chosen.

- At the chosen vertex, the fracture event occurs at breaking points.

- The number of vertices is counted and the area is computed using the *Shoelace formula* [58] of a $n$-side polygon,

$$\mathscr{A} = \frac{1}{2} \left| \sum_{i=0}^{n-1} \left( x_i y_{i+1} - x_{i+1} y_i \right) \right|. \tag{1}$$

- Repeat steps until all four vertices are fractured.

The code for this algorithm is available in the reference [59]. Fig 3 shows, as an example, the result of the numerical simulation for a square polygon with a side length of one.

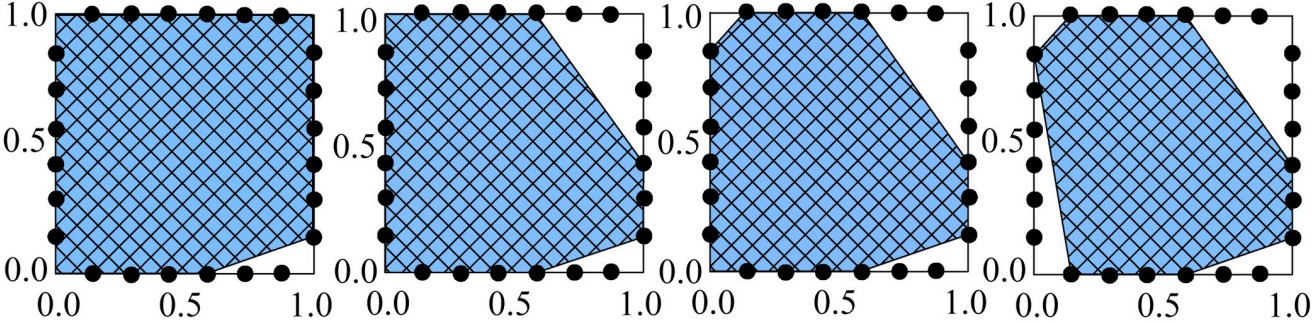

**Fig 3. An example of the evolution of a square polygon with six breaking points per side of length one.**

## 2.3 Numerical results

To answer the question about the most likely polygon that results after full fracturing, let us suppose we have a statistical ensemble (copies) of $M \to \infty$ square polygons (side length 1), which evolve in the same way (algorithm) as in Fig 3. After the evolution, each resulting polygon will have different areas and number of vertices. The probability distribution is built using this ensemble, and the most probable polygon will be the one with the highest frequency or repetition. In this sense, the numerical experiment was carried out using a square and another rectangular shape as initial polygons. The two had six breaking points, evenly distributed on each of their edges. Our results show that the probability distribution for the number of vertices is the same in both cases, namely, there is no difference if we use a square or a rectangle. This happens only when we focus solely on its number of vertices. Here, for the sake of simplicity, we show only the results for the square with a side length unity.

Fig 4 shows the frequency distribution as a function of the area and the number of vertices for squares with six possible breaking points on each side. The dataset corresponding to this distribution, and generated by the code is available in the S1 Dataset. Note in Fig 4 that the most probable resulting polygon will have 6 or 7 vertices and an approximate area of 0.56 to 0.66. As an application of our model, we shall study the breakages of the Rosetta Stone. Accordingly, the following sections are devoted to this task. The lost area of this stele is at least 16% (next section), which is consistent with the lost area of a square with 6 breaking points. In this sense, for our study, we will use the frequency distribution of Fig 4.

## 3 The Rosetta Stone as an example

As mentioned in the introduction, this stele has played an important role in understanding the writings of ancient Egyptian civilization. Fig 5 shows a schematic of the outline of the Rosetta Stone. Six vertices have been considered and related to the points of greatest breaks. Further,

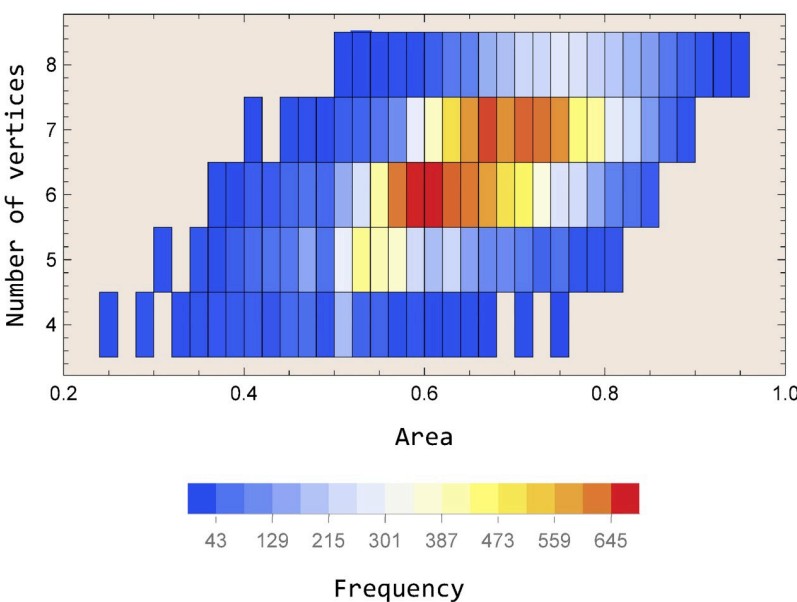

**Fig 4. For a square with six possible breaking points on each side, the frequency distribution as a function of the area and number of vertices.** Each bin has dimensions 0.02×1.

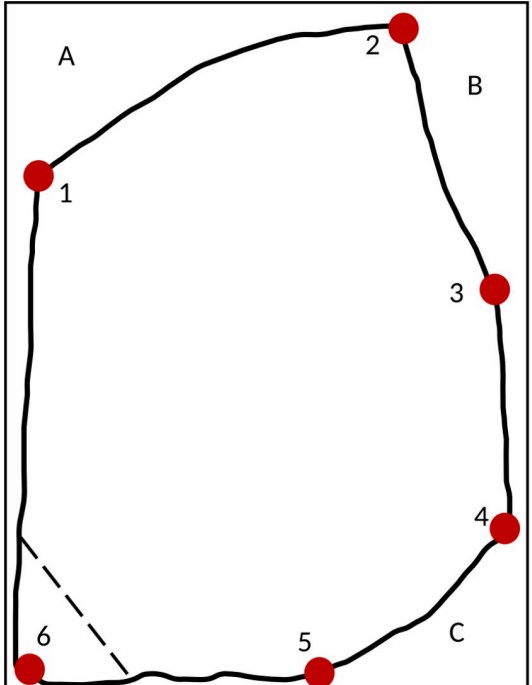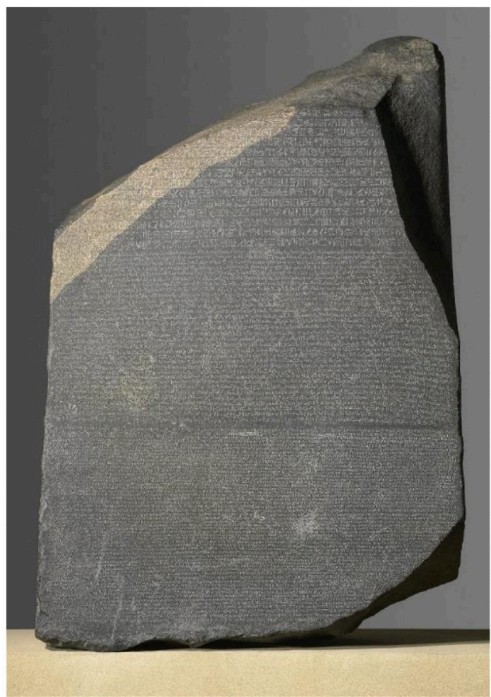

**Fig 5. A scheme of the face of the Rosetta Stone framed in a rectangle.** For simplicity, this rectangle is assumed the original piece.

the volumes lost have been denoted with A, B, and C. On the website, [60] a simulation to visualize this artifact in three dimensions is available.

For quantitative purposes, this section will use the image processing software *Image-J* [61] and the corresponding units of length will be in pixels. In this way, different geometric and statistical aspects related to this artifact will be quantified. In addition, arguments are given to consider possible new fractures.

In this context, in Fig 5, the dark red dots represent the six largest vertices of the current polygon. Vertex 5 (and slightly 6) is relatively smooth and the others are sharp. In the process of these fractures, we assume that the face of the Rosetta Stone went from 4 to 6 vertices. In fact, and importantly, six is the number of more probable vertices in the rupture process (section II. C). However, seven vertices also have a high probability. The angles associated with each vertex, starting with the number one, are 139˚; 112˚; 154˚; 141˚; 158˚, and 88˚ approximately. In the lower left, dotted line, an eventual new break. There is a probability (0.34) of seven vertices and four fractures (see below).

In Fig 5, the rectangular frame has dimensions of $\Delta x \sim 290$ and $\Delta y \sim 370$ which gives a perimeter of $\sim 1320$. The perimeter of the inscribed stone $\sim 1179$. Regarding the area, the rectangle has $\sim 107300$ while the stone has $\sim 90190$ (see Table 1).

**Table 1. Comparison between Rosetta Stone and Rectangle figure simulated.**

| Object | Area | Perimeter |
|---|---|---|
| Rosetta Stone | 90190 | 1179 |
| Rectangle | 107300 | 1320 |

**Table 2. Probability as a function of number of vertices.**

| Number of vertices | Probability |
|---|---|
| 4 | 0.03 |
| 5 | 0.16 |
| 6 | 0.35 |
| 7 | 0.34 |
| 8 | 0.12 |

The percentage of lost area is, at least, 16% (assumed as lost information). Moreover, that of the lost perimeter is at least, 11%.

As theoretically noted in section 1, in general, the number of vertices in the fracture process could range from 4 to 8 (see Table 2). Then, using the frequency distribution of Fig 4 in section II, 20 000 realizations, the corresponding probabilities of occurrence of these vertices will be: consequently, the face of the Rosetta Stone currently meets the maximum corresponding to six vertices and fracturing in three corners (areas A, B, and C). Nevertheless, the probability of a new fracture (7 vertices) is not low, being of the order of 0.34 (Fig 6). In this sense, since the most acute angle corresponds to vertex 6 (with 88˚). It can be conjectured that an eventual new fracture would occur in that region (vertex 6, cut line, Fig 5).

## 4 Griffith's criterion for fractures: Breaks sequence

On a large scale, the Rosetta Stone was fractured into three sections (A, B, and C). In addition, it has minor fractures that we do not consider in this research. Particularly, on the left side of the face, there are small losses of material that seem related to some kind of dragging.

In general, the fracture condition of an object requires that the stress-energy of the bulk be greater than the surface energy of the new area, which is related to the breaking of molecular bonds. That is, if the volume $V$ of the object and the surface $S$ of an eventual fissure comply

$$\frac{1}{2}\alpha\sigma^2 V \geqslant \gamma S, \tag{2}$$

then there will be fragmentation (Griffith criterion [21–23]).

In the previous inequality, $\alpha$ is a parameter proportional to the rigidity modulus. The parameter $\gamma$ is the energy per unit area to perform the crack and, finally, $\sigma$ is the average over the directions of the stress tensor [21, 25, 26]. The variable $\sigma$ has pressure units.

From inequality (2) it can be inferred that, as the volume decreases, the fractured surfaces will be smaller. Thus, based on the above inequality, it seems reasonable to conjecture that the temporal order of the breaks in the Rosetta Stone was: first section A, then section B and, later, section C. Note that, as a mentioned conjecture, we do not discard an eventual fourth break on point 6 (probability 0.34, Fig 6). This breakdown would necessarily entail a smaller lost area than the previous ones.

Considering points 1 to 5 as vertices (Fig 5), the corresponding estimates for the missing areas are (*Image-J*):

$$A \quad \sim 6700$$
$$B \quad \sim 6200$$
$$C \quad \sim 4300$$
$$X_6 \quad \sim 3400.$$

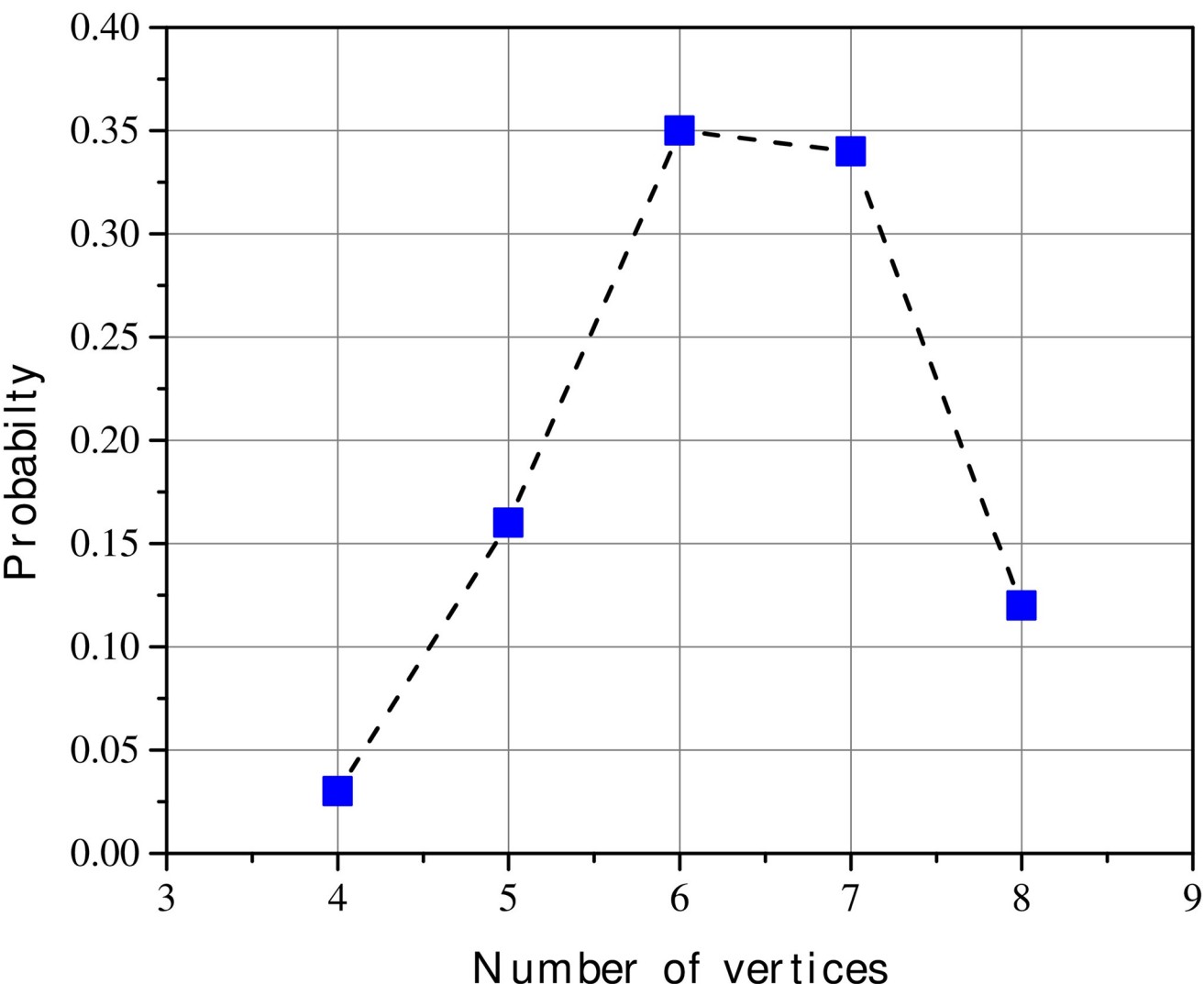

**Fig 6. Probability of obtaining a determined number of vertices due to fractures.** The Rosetta Stone currently has six vertices that match the maximum probability (0.35). However, also with high probability, it could have seven vertices. That is, there is a 0.34 probability that a new breakdown will appear.

Where the lost area $X_6$, of an eventual fourth fracture, has been incorporated by making a linear projection (Fig 5, lower left corner).

Additionally, the posterior face of the stone has five vertices. Point six has also almost a right angle on that back face. In the same way, an eventual fracture would occur there (probability 0.35, six vertices, Fig 6).

## 5 Loss of information: Entropy production

In this section, we consider the quantification of information loss, and the tool we will use is entropy. This concept is an essential part of thermodynamics, statistical mechanics [62–67] and information theory [68]. For a physical system at a given energy, the number of states accessible to the system (degeneracy) is related to the volume $V$ occupied by the system in phase space. Important, for our purposes, the phase space includes the physical space. Entropy in general terms is defined by the standard relation $S \propto \ln(V)$ due to Boltzmann.

As a solid stele loses fragments with information, such as the Rosetta Stone, entropy grows (irreversibility). From a quantitative point of view, this is very similar to Joule expansion in thermodynamics. Assuming that at a given instant the area loss is $A_L$ (dimensionless) then the entropy $S$ is given by:

$$S = \ln(1 + A_L), \tag{3}$$

where the initial missing area is assumed to be $A_L = 0$. Section IV considered the loss area (data) in the case of the Rosetta stele. Formally, this allows us to evaluate the sequential variation of entropy.

Explicit calculations reveal an increase in entropy and therefore a lack of information in this stele (with $R^2 = 0.98$, that quantifies goodness of fit):

$$S = 0.816\ln(N) + 8.842, \tag{4}$$

being $N$ the increasing number of missing vertices (irreversibility). The above expression was constructed using the four values of the loss areas in section IV and, formally, $1 \leq N \leq 4$. However, as mentioned in the introduction, other stelae with more complete information have subsequently been found. Therefore, entropy has decreased because the information has been retrieved (Maxwell's demon).

## 6 Conclusions

In this work, we have developed a numerical-statistical model to predict the probability of ruptures in a laminar-shape solid (Figs 2 and 5). Starting with an initial geometry, we find the most likely configurations in the systematic process of ruptures. While it is true that we consider an initial geometry of a square, this methodology can be adapted to other parallelepipeds.

Given the conditions imposed in our model with four initial vertices, in the generic process of ruptures, the probabilities of reaching six and seven vertices were explicitly calculated (probabilities of 0.35 and 0.34 respectively). Currently, being our main application, the Rosetta Stone has six vertices (three fractures) where part of the assumption is that the surface of the stone changes from 4 vertices to 6 vertices. Then, according to our calculations, there is an acceptable probability of a new fracture (seven vertices) in this archaeological artifact. Future applications to other stelae or archaeological objects will be considered.

Using Griffith's criterion, for solids under stress, it was possible to infer the sequential history of the three Rosetta Stone fractures, even conjecturing the relative size of an eventual fourth rupture.

On the other hand, in this work, we have considered the entropy of the system from the point of view of missing information (fragment dispersion). This exhibits formal mathematical similarity to the Joule expansion in thermodynamics. In the case of the Rosetta Stone, this measurement is proportional to the area lost in the rupture process because that part always increases (section 4). Future applications to other archaeological objects will be considered.

## Supporting information

**S1 Dataset. Dataset for Fig 4.**
(TXT)

## Acknowledgments

We appreciate the discussions with Veronica Bahoz and Angel R. Plastino, which inspired part of the motivation of this and future works.

## Author Contributions

**Conceptualization:** Marcelo Calcina-Nogales, Boris Atenas, Juan Cesar Flores.

**Formal analysis:** Marcelo Calcina-Nogales, Juan Cesar Flores.

**Investigation:** Marcelo Calcina-Nogales, Juan Cesar Flores.

**Methodology:** Marcelo Calcina-Nogales, Juan Cesar Flores.

**Software:** Marcelo Calcina-Nogales, Juan Cesar Flores.

**Writing – review & editing:** Marcelo Calcina-Nogales, Boris Atenas, Juan Cesar Flores.

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
