## [Decision Letter · Decision Letter 0]

7 Aug 2023

PONE-D-23-12788Griffith theory of physical fractures, statistical procedures and entropy  production: Rosetta stone’s legacyPLOS ONE

Dear Dr. Atenas,

Thank you for submitting your manuscript to PLOS ONE. After careful consideration, we feel that it has merit but does not fully meet PLOS ONE’s publication criteria as it currently stands. Therefore, we invite you to submit a revised version of the manuscript that addresses the points raised during the review process.

Please, address all the comments made by the reviewer. 

We look forward to receiving your revised manuscript.

Kind regards,

Antonio Riveiro Rodríguez, PhD

Academic Editor

PLOS ONE

Journal Requirements:

2 Please note that PLOS ONE has specific guidelines on code sharing for submissions in which author-generated code underpins the findings in the manuscript. In these cases, all author-generated code must be made available without restrictions upon publication of the work. Please review our guidelines at https://journals.plos.org/plosone/s/materials-and-software-sharing#loc-sharing-code and ensure that your code is shared in a way that follows best practice and facilitates reproducibility and reuse.

"No."

"NO authors have competing interests"

Reviewers' comments:

Reviewer's Responses to Questions

**Comments to the Author**

1. Is the manuscript technically sound, and do the data support the conclusions?

Reviewer #1: Yes

2. Has the statistical analysis been performed appropriately and rigorously? 

Reviewer #1: Yes

3. Have the authors made all data underlying the findings in their manuscript fully available?

Reviewer #1: Yes

4. Is the manuscript presented in an intelligible fashion and written in standard English?

Reviewer #1: Yes

5. Review Comments to the Author

Reviewer #1: 1. The introduction part introduces too much about the stele itself and too little about the experimental process and research direction of this paper. It is suggested to adjust the proportion of the introduction part to highlight the theme.

2. The design part of the model uses a square model with a side length of 2cm, while the actual stone tablet was a rectangle before it was broken. It is suggested to add a contrastive description of square and rectangle in the model design to improve the rigor of the experiment.

3. In the design of the model, the numerical result directly illustrates the same loss area as the square with 6 breakpoints. It is suggested to add 5 or other breakpoints corresponding to the data and stone tablet actual data comparison process.

4. 2 Take the Rosetta Stone as an example. Part of the assumption is that the surface of the stone changes from 4 vertices to 6 vertices.

5. It is suggested that the definition of entropy should be added before the formula of entropy, so as to facilitate readers' understanding of this part.

6. The conclusion is suggested to be rewritten. The conclusion of this paper is the description and summary of the previous experimental results, without innovation and sublimation part of the description.

7. There are obvious problems with the format of the chart. You need to carefully check the format of each icon and correct formatting errors in the chart.

8. There are some basic grammatical errors in the article.

9. The original formula in the article is suggested to supplement the formula derivation process.

10. There are obvious errors in the structure of the article. It is recommended to recalibrate the structure of the article and indent the first line at the beginning of part 3.

11. At the end of the paper, the suggestions of future outlook are described, for example, the improvement ideas of the model and the process of data processing are described in detail.

12. The contribution description of the article is not obvious enough, so it is suggested to supplement the detailed contribution of the article.

6. PLOS authors have the option to publish the peer review history of their article (what does this mean?). If published, this will include your full peer review and any attached files.

Reviewer #1: No

---

## [Author Response · Author response to Decision Letter 0]

18 Sep 2023

The response to the Academic Editor and Reviewer has been appended within the document titled 'Response to Reviewers. We add a copy here.

Response to Editor:

1) We formatted the article according to PLOS ONE’s style requirements. 

2) The generated code (written in Mathematica) is available in the following repository https://doi.org/10.5281/zenodo.8317810 .

3) The authors received no specific funding for this work.

4) The authors have declared that no competing interests exist.

5) The minimal underlying dataset is generated by the code that outputs a txt file (S1_Dataset.txt). An example of the txt file was uploaded as supporting information to PLOS ONE.

6) The provided code (https://doi.org/10.5281/zenodo.8317810) outputs the minimal underlying data set as a txt file. The txt file (S1_Dataset.txt) was uploaded as supporting information to PLOS ONE.

7) We have reviewed our reference list, and it is complete and correct.

Response to Reviewer

1) Thank you for pointing this out. We added a new paragraph that explains more about the experimental process and shrunk the paragraphs related to the stele.

2) We added a paragraph that contrasts the probability distribution of a square and a rectangle.

3) Thanks for the suggestion. Certainly, the actual simulation is set to six breaking points, whereas our illustrative Figures 2 and 3 have only five breaking points. We have changed the respective figures and set them to six breaking points.

4) Thanks for the comment. We have added your suggestion to our article (Section 5, paragraph 2).

5) Thank you for this suggestion. Accordingly, we added the definition of entropy before equation 3. 

6) We thank the referee for this comment. Effectively revised the conclusions, we have rewritten the terms indicated (emphasizing innovation, important and novel aspects developed in our work). The conclusions have also been written promoting more generic aspects.

7) We have, accordingly, changed the format of the chart in figures 4 and 6.

8) Thank you for pointing this out. We went through the entire manuscript to eliminate grammatical errors.

9) This comment of the referee was combined with comment 5. And that formula was better explained. Thank you for this suggestion. In the same sense, we have given greater support to the rule of fractures related to the Griffiths criterion by incorporating explicit references to that theory. 

10) We have adjusted the article's formatting to align with the PLOS ONE template. We took great care to meticulously review and restructure the entire article, paying close attention to the indentation at the commencement of each section. Your guidance has proven immensely helpful in enhancing the overall quality and presentation of the manuscript.

11) Thank you for pointing this out. We have considered your comment, and it was incorporated with comment 6 in the conclusions section.

12) Thank you for this suggestion. We have reinforced the contribution of the article in the conclusion section. Some applications of the model are described in the Introduction section (see the paragraphs Rolling stones in rivers, Crack networks, and Breakage of solid structures).

---

## [Decision Letter · Decision Letter 1]

21 Sep 2023

Griffith theory of physical fractures, statistical procedures and entropy  production: Rosetta stone’s legacy

PONE-D-23-12788R1

Dear Dr. Atenas,

We’re pleased to inform you that your manuscript has been judged scientifically suitable for publication and will be formally accepted for publication once it meets all outstanding technical requirements.

Kind regards,

Antonio Riveiro Rodríguez, PhD

Academic Editor

PLOS ONE

Reviewers' comments:

Reviewer's Responses to Questions

**Comments to the Author**

1. If the authors have adequately addressed your comments raised in a previous round of review and you feel that this manuscript is now acceptable for publication, you may indicate that here to bypass the “Comments to the Author” section, enter your conflict of interest statement in the “Confidential to Editor” section, and submit your "Accept" recommendation.

Reviewer #1: All comments have been addressed

2. Is the manuscript technically sound, and do the data support the conclusions?

Reviewer #1: Yes

3. Has the statistical analysis been performed appropriately and rigorously? 

Reviewer #1: Yes

4. Have the authors made all data underlying the findings in their manuscript fully available?

Reviewer #1: Yes

5. Is the manuscript presented in an intelligible fashion and written in standard English?

Reviewer #1: Yes

6. Review Comments to the Author

Reviewer #1: The article has made significant improvements. All comments have been answered. It can be accepted as it is.

7. PLOS authors have the option to publish the peer review history of their article (what does this mean?). If published, this will include your full peer review and any attached files.

Reviewer #1: No

---

## [Editor Report · Acceptance letter]

27 Sep 2023

PONE-D-23-12788R1 

Griffith theory of physical fractures, statistical procedures and entropy  production: Rosetta stone’s legacy 

Dear Dr. Atenas:

I'm pleased to inform you that your manuscript has been deemed suitable for publication in PLOS ONE. Congratulations! Your manuscript is now with our production department. 

Kind regards, 

on behalf of

Dr. Antonio Riveiro Rodríguez 

Academic Editor

PLOS ONE